# Thermally Crosslinked Hydrogen-Bonded Organic Framework Membranes for Highly Selective Ion Separation

**DOI:** 10.3390/molecules28052173

**Published:** 2023-02-26

**Authors:** Xiyu Song, Chen Wang, Xiangyu Gao, Yao Wang, Rui Xu, Jian Wang, Peng Li

**Affiliations:** 1School of Physical Education, Shanghai University of Sport, Shanghai 200438, China; 2Department of Chemistry and Shanghai Key Laboratory of Molecular Catalysis and Innovative Materials, Fudan University, Shanghai 200437, China

**Keywords:** hydrogen-bonded organic framework (HOF), thermally crosslinked, ion separation

## Abstract

The weak bonding energy and flexibility of hydrogen bonds can hinder the long-term use of hydrogen-bonded organic framework (HOF) materials under harsh conditions. Here we invented a thermal-crosslinking method to form polymer materials based on a diamino triazine (DAT) HOF (FDU-HOF-1), containing high-density hydrogen bonding of N-H⋯N. With the increase of temperature to 648 K, the formation of –NH– bonds between neighboring HOF tectons by releasing NH_3_ was observed based on the disappearance of the characteristic peaks of amino groups on FDU-HOF-1 in the Fourier transform infrared (FTIR) and solid-state nuclear magnetic resonance (ss-NMR). The variable temperature PXRD indicated the formation of a new peak at 13.2° in addition to the preservation of the original diffraction peaks of FDU-HOF-1. The water adsorption, acid-base stability (12 M HCl to 20 M NaOH) and solubility experiments concluded that the thermally crosslinked HOFs (TC-HOFs) are highly stable. The membranes fabricated by TC-HOF demonstrate the permeation rate of K^+^ ions as high as 270 mmol m^−2^ h^−1^ as well as high selectivity of K^+^/Mg^2+^ (50) and Na^+^/Mg^2+^ (40), which was comparable to Nafion membranes. This study provides guidance for the future design of highly stable crystalline polymer materials based on HOFs.

## 1. Introduction

Emerging porous materials constructed by the design rules of reticular chemistry, such as metal-organic frameworks (MOFs) [1,2], covalent organic frameworks (COFs) [3,4], and hydrogen-bonded organic frameworks (HOFs) [5,6,7,8], are crystalline porous materials with atomic-level precision and have a wide range of applications in gas storage and separation, sensing, and catalysis due to their permanent porosity and designability of structure and function [9,10,11]. Unlike the coordination bonds in MOFs and covalent bonds in COFs, HOFs are composed of organic or metallic structured building blocks through intermolecular H-bonding interactions [12]. HOFs have been rapidly evolving into an important and unique class of functional porous materials that exhibit excellent solution processability, gentle preparation, and easy regeneration and recycling. The term HOF was created in 2011 by Chen et al., who reported the first HOF with a 2,4-diaminotriazine groups (DAT) (i.e., HOF-1) that was used for the separation of C_2_H_2_/C_2_H_4_ [13,14]. Chen’s coining of the term HOFs marked the beginning of this field and greatly stimulated interest in exploring new HOFs for various applications. Since then, the practice of hydrogen bonding chemistry for extended networks has led to many HOFs with high porosity and Brunauer–Emmett–Teller (BET) surface area. Rational combination of molecular backbone and H-bonding units has led to the formation of robust and porous HOFs with high porosity. Since then, a series of DAT-based HOFs [15,16,17,18,19,20,21,22,23] have been prepared and show diverse applications. The DAT group has three ways to form H-bonded dimers: head-to-head, side-by-side, and head-to-side (Figure 1). Their diversity is due to the presence of excess hydrogen bond donors and acceptors. Because of this, the H-bonding network can expand in more dimensions, while solvent molecules may be able to have additional hydrogen binding sites which occupy the pore space. It is a formidable challenge to construct permanently porous HOFs materials based on organic units of DAT.

The weak H-bonding energy (10–40 kJ mol^−1^) and the flexibility of hydrogen bonds make HOFs extremely fragile and most of them are prone to collapse after the removal of solvent molecules by heat treatment or vacuum activation. Therefore, the construction of permanently porous HOFs is an important challenge. So far, for example, (1) π-π stacking interactions are attractive and exhibit non-covalent interaction between aromatic rings [18,24,25,26,27,28,29,30,31,32]; (2) interpenetrated frameworks, in which two or more frameworks with the same composition are locked together, provide a more robust framework [33,34,35,36,37]. Charge-assisted H-bonding interactions [38,39] and cross-linking modification [40] strategies have been proposed to further improve the stability of HOFs.

Based on the above strategies, a series of HOFs materials with permanent porosity were synthesized using various organic units with high-density hydrogen bond donors and acceptors. Recently, Ke et al. have enhanced the chemical and structural stability by covalently crosslinking hydrogen-bonded, pre-organized molecular crystals with covalent bonds to strengthen the hydrogen-bonded network, which has led to further insights in the design of HOFs materials [41,42]. In addition to the chemical crosslinking method, however, the thermal crosslinking method is seldom used to prepare polymer materials from HOFs. Considering the potential advantages of simple synthesis, easy processing, and scaled-up preparation, it is highly desirable to develop a novel synthesis method to obtain thermally crosslinked HOF materials. In a sense, thermally crosslinked HOFs are also similar to COF materials.

Membrane separation is a popular technology with the advantages of high separation efficiency and low energy consumption [43]. Polymer membranes currently dominate the commercial market; however, it is difficult to control the pore structure at the molecular level, and their drawbacks are more prominent when used for precise ion separation [44]. In addition, the separation performance of polymer membranes usually receives Robeson upper limits [45], resulting in a poor balance between permeability and selectivity. However, HOF materials have the advantages of controlled pore structure and adjustable ion-specific functional groups, making them novel candidates for the development of membrane separation technologies. Unlike desalination processes that block various ions, ion separation selectively permeates the smaller size and lower valence ions but blocks larger ions, which is critical in the battery and mining industries, such as the recovery of lithium ions and other precious metal ions [46]. Achieving accurate ion separation remains a great challenge and there is an urgent need for HOF membrane materials with high selectivity to achieve ion separation sieving.

As key starting materials of graphite-like carbon nitride (g-C_3_N_4_), melamine and its derivatives have been generally used to synthesize carbon-nitride-based polymers (CNP) by direct high-temperature calcination, during which ammonia is released, and -NH-covalent bonds are formed. CNP is widely used in many applications for its ideal visible light response, environmental and low-cost advantages, and tunable electronic properties [47,48]. CNP materials generally cannot be dissolved in acids, bases, or organic solvents and are very thermally stable in air; however, they have a low BET surface area, and many are also non-porous, which limits the application of this material. Melamine and HOF-8 with DAT groups can be carbonized at high temperature (973 K to 1273 K) to obtain metal-free microporous nitrogen-doped carbon materials, which reduces the disadvantages of low BET and low nitrogen content of the original carbon materials and forms carbon materials with uniform active sites and high BET [49,50]. We speculate that DAT-functionalized HOFs may also cross-link under heating to form highly stable porous polymer materials (Figure 1). Here, we report an example of DAT-based HOF (FDU-HOF-1) material that could be thermally crosslinked with itself and neighboring molecules into partially covalently bonded organic frameworks.

With this strategy, the thermally crosslinked HOF materials retain high crystallinity and porosity, while showing significantly improved chemical stabilities under strong acid (12 M HCl) and base conditions (20 M NaOH) for three days. As a proof of functionality, the thin-film membranes fabricated with the thermally crosslinked HOF materials demonstrate high ion permeability (up to 270 mmol m^−2^ h^−1^ for K^+^) and K^+^/Mg^2+^ selectivity (up to 50), which is even comparable to commercial Nafion membranes.

## 2. Results and Discussion

6-[4-(4,6-diamino-1,3,5-triazin-2-yl)phenyl]-1,3,5-triazine-2,4-diamine (BDAT) was synthesized following the previous procedures [51]. Briefly, it is the reaction of dicyanobenzene and dicyandiamide in N’N-Dimethylformamide (DMF). BDAT was dissolved in DMF in an uncapped vial. Then this vial was placed in a bigger vial containing water. Following this, it was placed in a baking oven for about a week to obtain large crystals of FDU-HOF-1. Single-crystal X-ray diffraction (SCXRD) revealed FDU-HOF-1 in an orthorhombic P*nnm* space group (a = 3.6813 Å, b = 11.5900 Å, c = 15.8935 Å, α = β = γ = 90°). In FDU-HOF-1, each BDAT molecule is interconnected with eight adjacent units using N-H···N hydrogen bonds to form a two-dimensional (2D) framework. These 2D layers are stacked in an ABAB pattern through π-π stacking interactions, forming 3D structures with a one-dimensional pore channel of 4.7 Å × 9.5 Å along the [101] direction (Figure 2b). The calculated different free spaces of FDU-HOF-1 are 20% (564.07 Å ^3^ as void volume, Connolly radius 0.6 Å), 14% (593.35 Å^3^ as void volume, Connolly radius 0.9 Å), and 10% (616.85 Å as void volume, Connolly radius 1.3 Å) (Appendix A). The porosity of FDU-HOF-1 was initially verified by calculation, indicating that FDU-HOF-1 is a porous HOF material.

The powder X-ray diffraction (PXRD) patterns of FDU-HOF-1 were closely matched with the simulated one. To further test the thermal stability of FDU-HOF-1, we measured the variable temperature PXRD (VT-PXRD) (Figure 3a). It could be seen by PXRD that at higher than 373 K, FDU-HOF-1 still maintains high crystallinity, while the peak does not change, indicating that FDU-HOF-1 was a stable HOF material. We named FDU-HOF-1 at different temperatures as FDU-HOF-1-X (X = 298 K, 573 K, 623 K, 648 K, and 673 K). We found that the PXRD pattern of FDU-HOF-1 basically remained at high crystallinity with no change until 523 K, and some peaks disappeared at 648 K. When heated to 673 K, its PXRD diffraction intensity becomes weaker, and its peaks basically all disappear when it reaches 700 K. This indicates that its thermal stability cannot exceed 700 K. When it exceeds 700 K, it loses its crystallinity and is no longer an ordered HOF material. We observed that the powder color of FDU-HOF-1 gradually deepened with the increase in temperature (Figure 3b). To further understand the changes, we measured the Fourier transform infrared (FTIR) spectroscopy of FDU-HOF-1-X (298 K, 573 K, 623 K, and 673 K) (Figure 3c). In the FTIR spectra, FDU-HOF-1-298 K and FDU-HOF-1-573 K treatments still had the characteristic absorption double peak of intermolecular amino hydrogen bonds at 3500–3300 cm^−1^, which was absent at 623 K and 673 K treatments, indicating that the original N-H···N hydrogen bond disappeared. To better interpret the materials of HOFs after thermal cross-linking, we tested the solid NMR of FDU-HOF-1-298K and FDU-HOF-1-648K. The Solid-state 1 H NMR spectrum experiments showed a great correlation with the FIIR for the solid materials. A new peak was generated at about 1 ppm, which we presume to be the peak of hydrogen on -NH- after thermal crosslinking (Appendix A). We purposely studied the effect of heating time on the phase change of FDU-HOF-1. As shown in Figure 3d, we compared FDU-HOF-1-298 K and FDU-HOF-1-648K-x (x = 12, 24, 36 h); the PXRD peaks of the material changed at 648 K. The peaks at 11°, 15.1°, and 18.3° all fade away. After heating FDU-HOF-1-648K to 36 h, there was only about 13.2° of the new peak, the rest of the peak (11°, 15.1°, and 18.3°) disappeared. We consider the peak at about 13.2° as the characteristic peak after thermal crosslinking. Therefore, we believe that FDU-HOF-1-648 K, the N-H···N hydrogen bond in FDU-HOF-1 completely disappears and gradually thermally crosslinks into covalent bonds of –NH–. We also examined the powder color of FDU-HOF-1-648K at 12 h, 24 h, and 36 h, respectively, and found that the material color gradually became darker yellow (Appendix A).

To further investigate the stability of the material, we dissolved FDU-HOF-1 and thermally crosslinked samples in DMF solution (Figure 4a). With the increase in temperature and heating time, the solubility of the formed material becomes worse and worse until it becomes completely insoluble at 673 K treatment. This also proves that our idea is correct, i.e., the solubility then decreases, and the material is gradually thermally crosslinked into covalent bonds. We perform further stability tests on FDU-HOF-1 processed under 648 K conditions. Notably, as shown in Figure 4b, FDU-HOF-1-648K maintained its crystallinity and stability under harsh alkaline conditions (20 M NaOH) and acidic conditions (1 M HCl); however, its crystallinity decreased under strongly acidic conditions (12 M HCl). Many stable HOFs are unstable in alkaline solutions because they undergo deprotonation or hydrolysis. For example, Trispyrazole-1 (1 M HCl and 2 M NaOH for 30 days) [52], CPHAT-1a (10% HCl for 24 h) [53], HOF-101/PFC-1 (12 M HCl for 117 days) [26], HOF-16a (pH 1~7 for 7 days) [54], HOF-FAFU-1 (pH 1~9 for 12 h) [55], PFC-5 (0.1 M HCl for 24 h) [56], HOF-76a (pH 1~10 for 24 h) [57], HOF-110 (pH 1~7 for 16 h) [58], BPMCz-P1 (10% HCl for 48 h) [59], HOF-25 (pH 7~11 for 7 days) [60], CBPHAT-1a (37% HCl for 7 days) [61], and HOF-40 (pH 1~14 for 5 days) [62]. The chemical stability of most HOFs materials is not too good; some HOFs are only acid resistant, some HOFs are only alkali resistant, and only a few are stable under both acid and alkali conditions. These are made possible by its stable build model. In contrast, FDU-HOF-1-648K exhibits excellent chemical stability, thanks to its heating crosslinking to form more stable covalent bonds and multiple π-π intermolecular interactions, superior to previously reported HOFs materials with DAT.

Many HOFs have been discovered to have excellent thermal and chemical stability; however, most of them are unable to maintain the skeletal structure during water adsorption, resulting in skeleton collapse. To further elucidate the water stability and porosity after thermal crosslinking, the water vapor sorption of FDU-HOF-1-298K and FDU-HOF-1-648K-x (x = 12, 24, 36 h) at 298 K were collected to probe the structural stability line. As shown in Figure 5, it is found that FDU-HOF-1-298K and FDU-HOF-1-648K-x (x = 12, 24, 36 h) have S-shaped sorption isotherm, and the water absorption gradually increases until 48% RH, and then suddenly absorbs until 72% RH. FDU-HOF-1-298K and FDU-HOF-1-648K-x (x = 12, 24, 36 h) adsorb 13.7, 16.7, 19.6, and 20.8 wt% water vapor at 298 K, respectively. Surprisingly, the thermally crosslinked HOFs adsorb even more water vapor than the pristine FDU-HOF-1, and the amount of water absorbed rises as the degree of thermal crosslinking increases in the HOFs. This indicates that the pores are filled with water, which proves its high-water stability and that it maintains its porosity. The excellent water stability has surpassed most HOFs materials, as well as many MOFs and COFs materials, which indicates that thermally crosslinked FDU-HOF-1 (TC-HOF) has great potential to be applied in practical situations.

Encouraged by the high thermal/chemical stabilities and water uptake ability of the FDU-HOF-1 and their thermally crosslinked derivative materials, we moved on to preparing HOF-based thin films. The thin films of FDU-HOF-1 (~3 μm in thickness) were prepared by spin-coating a dilute monomer solution of DMF onto a porous silica membrane support and were dried under flowing dry air at room temperature (Figure 6a–c). The film was further heated at 648 K for different times (12 h, 24 h, 36 h) in N_2_ atmosphere to form thermally crosslinked HOF thin films (noted as TC-HOF-TF-x, x = 12, 24, 36). The scanning electron microscopy (SEM) images showed the densely packed HOF particles and defect-free cross-section of the HOF thin films (Figure 6d–f). The successful preparation of the films also prompted us to further explore their application properties.

Permeation selectivity of the FDU-HOF-1/silica and TC-HOF-TF/silica membrane were investigated with a two-chamber cell, and selective ion transport was demonstrated using concentration-driven diffusion experiments (Figure 7). The ion permeation rate of FDU-HOF-1 membranes was much lower than that of TC-HOF-TF membranes. The permeation rate of K^+^ ions is in the order of 201 < 221 < 247 < 270 mmol m^−2^ h^−1^ for FDU-HOF-1 and TC-HOF-TF-x (x = 12, 24, 36), respectively. The K^+^ ions permeation rate of TC-HOF-TF-36 was higher than that of DMBP-TB (200 mmol m^−2^ h^−1^) and PIM-1 (5 mmol m^−2^ h^−1^) [63]. For Na^+^, Li^+^, and Mg^2+^, the trends of permeation rate are similar for FDU-HOF-1 and TC-HOF-TF. The faster diffusion rate in TC-HOF-TF-x over FDU-HOF-1 is possibly attributed to their higher porosity, which is consistent with the observation in water vapor adsorption experiments. However, the FDU-HOF-1 membrane shows a better ion selectivity, which allows the transport of smaller ions (K^+^, Na^+^, Li^+^ with 0.125, 0.184, 0.238 nm in Stokes radius), while rejecting larger Mg^2+^ (0.347 nm in Stokes radius). Both the FDU-HOF-1 membrane and TC-HOF-TFs show the selectivity of K^+^/Mg^2+^ (up to 30–50) and Na^+^/Mg^2+^ (up to 30–40). The selectivity of K^+^/Mg^2+^ is higher than that of many membrane materials, such as PIM-TA-TB (K^+^/Mg^2+^ selectivity 13.6) and AO-PIM-1 (K^+^/Mg^2+^ selectivity 33), but lower than PIM-BzMA-TB (K^+^/Mg^2+^ selectivity 93) and DMBP-TB (K^+^/Mg^2+^ selectivity 140) [63]. To the best of our knowledge, the permeation rate and selectivity of these HOF and TC-HOF-TF membranes are even comparable to that of commercial Nafion membranes, which have the potential for a range of applications, such as ion separation, and wastewater treatment.

## 3. Materials and Methods

### 3.1. Materials

1,4-dicyanobenzene, dicyandiamide, and potassium hydroxide (KOH) were supplied by Shanghai Titan Technology Co., Ltd. (Shanghai, China). Sodium hydroxide (NaOH), potassium chloride (KCl), sodium chloride (NaCl), lithium chloride (LiCl), magnesium chloride (MgCl_2_), and hydrochloric acid (HCl) were supplied by Adamas (Shanghai, China). N’N-Dimethylformamide, methanol, ethanol, and acetone were supplied by Sigma-Aldrich and Aladdin (Shanghai, China). Unless otherwise indicated, all chemicals and solvents used for synthesis were purchased from commercial sources and used as received without any further purification.

#### 3.1.1. Synthesis of BDAT

6-[4-(4,6-diamino-1,3,5-triazin-2-yl)phenyl]-1,3,5-triazine-2,4-diamine (BDAT) was synthesized following the previous procedures (Figure 2) [51]. 1,4-dicyanobenzene (1.544 g, 9.2 mmol), dicyandiamide (4.048 g, 48 mmol), and KOH (1.124 g, 20 mmol) into 200 mL of DMF. The two solutions were then mixed in a 250 mL round bottom flask and refluxed under a nitrogen atmosphere and stirred at 140 °C for 20 h. The product was washed several times with methanol and dried under vacuum at 70 °C to obtain the product BDAT, with a yield of about 85%. The molecular formula was C_12_H_12_N_10_. ^1^H NMR (600 MHz, DMSO): δ (ppm) 8.32 (s, 4 H), 6.82 (s, 8 H) (Appendix A).

#### 3.1.2. Synthesis of FDU-HOF-1 Single Crystals

10 mg of BDAT was added to a 5 mL vial with 1 mL DMF. Then, the 5 mL vial was put without a cover into a 20 mL vial filled with 5 mL water solution, and a lid was put on the 20 mL vial. After standing for a week or longer, the white single crystals were collected for single-crystal X-ray diffraction analysis.

#### 3.1.3. Synthesis of FDU-HOF-1 Powder

100 mg of BDAT was dissolved in 10 mL of DMF at 298 K to produce a clear solution. The solution was poured into 30 mL of water under stirring (300 rpm) within 1 min. The suspension was kept stirring for 6 h and isolated by centrifugation at 8000× *g* rpm for 5 min. The obtained white powder was further washed with acetone (2 ×45 mL), and then dried at room temperature (yield 90 mg, 90%).

#### 3.1.4. Synthesis of HOF Thin-Films and TC-HOF-TF

HOF thin films were fabricated by spin-coating dilute BDAT solutions (1 mg/mL in DMF) onto a porous silica membrane. BDAT solutions were filtered through syringe filters and dropped onto a porous silica membrane, and then the silica membrane was rotated at a speed of 1000 rpm with an acceleration speed of 500 rpm for 1 min. The DMF solvent was removed in vacuo at 90 °C for 8 h and dried under flowing dry air at room temperature for at least 24 h. The TC-HOF-TF-x (x = 12, 24, 36) membranes were prepared by loading the resulting HOF thin film into a tube furnace and was heated at 684 K for 12, 24, and 36 h, respectively.

### 3.2. Methods

#### 3.2.1. Powder X-Ray Diffraction

Powder X-ray diffraction was measured at room temperature on a STOE-STADI P powder diffractometer equipped with an asymmetric curved Germanium monochromator (CuKα1 radiation, λ = 1.54056 Å) and a one-dimensional silicon strip detector (MYTHEN2 1 K from DECTRIS). The line-focused Cu X-ray tube was operated at 40 kV and 40 mA.

#### 3.2.2. Variable Temperature PXRD (VT-PXRD)

Variable temperature PXRD measurements of HOFs were conducted on a STOE-STADI MP powder diffractometer operating at 40 kV voltage and 40 mA current with Mo-Kα1 X-ray radiation (λ = 0.71073 nm) in spinning capillaries in the temperature range of 20 to 400 °C under vacuum.

#### 3.2.3. Single-Crystal X-Ray Diffraction

Single crystals of FDU-HOF-1 were mounted on a Bruker D8 Venture MetalJet X-ray diffractometer equipped with a Photon II detector, and measurements of diffractions data were collected at 173 K. X-rays were generated by the Ga/In source (λ = 1.34138 Å) at 200 W (70 kV, 2.86 mA). Details of the crystal data, data collection, structure solution, and refinement are shown in Appendix A. CCDC 2,238,476 contain the crystallographic data for FDU-HOF-1, and the data can be obtained free of charge from The Cambridge Crystallographic Data Centre via www.ccdc.cam.ac.uk/structures (acceseed on 5 February 2023).

#### 3.2.4. NMR Measurements

Solution ^1^H NMR spectra were collected on a 500 MHz Bruker AVANCE III HD spectrometer at 298 K.

#### 3.2.5. Measurement of FTIR Spectra

The FTIR (Fourier transform infrared) spectra of powder samples were recorded in the 400–4000 cm^−1^ frequency region using a KBr discs method. FTIR spectra of nanofiber samples were using a slicing method.

#### 3.2.6. Solid-state ^1^H NMR Spectra

Solid-state NMR experiments were performed on a Bruker WB Avance II 400 MHz spectrometer.

#### 3.2.7. Chemical Stability Test

50 mg of HOFs powder was immersed in 30 mL of HCl (1 M and 12 M), NaOH (14 M and 20 M), and water (pH = 7) for 3 days and 7 days. The samples were then isolated by centrifugation and washed with water (3 ×45 mL) and acetone (3 × 45 mL) dried under vacuum at 80 °C for 12 h before PXRD test.

#### 3.2.8. Scanning Electron Microscope

Scanning electron microscope (SEM) images were obtained on a Phenom Prox microscope (Phenom Netherlands) at an acceleration voltage of 4.8 kV–15 kV.

#### 3.2.9. Water Vapor Adsorption

Water isotherms were measured on a Micromeritics 3Flex, and the water uptake in g g^−1^ units is calculated as [(adsorbed amount of water)/(amount of adsorbent)]. Prior to the water adsorption measurements, water (analyte) was flash frozen under liquid nitrogen and then evacuated under dynamic vacuum at least 3 times to remove any gases in the water reservoir. The measurement temperature was controlled with a Micromeritics temperature controller.

#### 3.2.10. Ion Permeability Test

The ion diffusion tests were carried out using stirred H-shaped cells. Membrane samples were sandwiched between two O-rings and sealed in the middle of the H-shaped cells. The effective area of the membrane samples in the H-shaped cell was 0.8 cm^2^. 30 mL of 5 mM salt solution (KCl, NaCl, LiCl, MgCl_2_) was used as the feed solution, and the permeate side was filled with 30 mL of deionized water (pH = 7). The ionic concentration of the permeate solution over time was measured by ICP-OES.

## 4. Conclusions

In summary, we synthesized a microporous HOF material FDU-HOF-1 based on DAT with high-density hydrogen bonding of N-H⋯N. After 36 h of treatment at 648 K, the HOF material could be converted into a thermally crosslinked polymer material TC-HOF. Notably, TC-HOF has excellent chemical stability (12 M HCl to 20 M NaOH), which exceeds most porous MOF and HOF materials. In addition, TC-HOF films have high ionic permeability and high K^+^/Mg^2+^ selectivity. This work provides a new strategy for the construction of novel stable crystalline porous polymer materials.

## Data Availability

The data presented in this study are available on request from the corresponding author.

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
