# Peer review of "Thermally Crosslinked Hydrogen-Bonded Organic Framework Membranes for Highly Selective Ion Separation"

_molecules, 2023, doi:10.3390/molecules28052173_

Round 1

Reviewer 1 Report

The paper presents an improtant and well-organized study of a new HOF. Adsorption properties as well as metal ion permeability were investigated. Thermal-crosslinking procedure has been systematically applied to the HOF samples, and it was shown to considerably increase their stability. Some inmprovement in water adsorption and metal ion permeability was also evidenced after the thermal crosslinking. 

In my opinion, the work obviously deserves to be published after considering some minor notes and corrections: 

1. Considering the mechanism of the thermal crosslinking, the product described by the authors can be deduced as just a covalent organic framework (COF). This is obviously not a crime to use rarer therms, but authors are suggested to highlight in the discussion, that the product of thermal crosslinking can be considered as COF as well. 

2. The summary formula reported in the cif file is apparently wrong, what leads to the faults in the formula-derived values, e.g. absorption coefficient (alert A in the checkcif). The cif must be corrected and resubmitted to CCDC. 

3. I regret to state that, according to PXRD (figure 4b), FDU-HOF-1-648 K-36h is not stable in 12M HCl, due to the disappearance of most peaks on the diffraction pattern after the treatment by HCl. Authors should soften the formulations concerning the stability of the samples in the acidic media. 

4. SEM images are shown in Figure 6, but only TEM instrumentation is desribed in the experimental part. Please clarify, what kind of technique has been used in the work. 

Reviewer 2 Report

The authors fabricated HOF materials based on diamino triazine containing high density of H-N…N hydrogen bonds. This HOF material could convert into a thermally crosslinked polymer TC-HOF. This material showed high ionic permeability of K+/Mg2+ ions that might be used in separation membranes.

(1) Figure 3, it is obvious that the phase changed from 298K to 673K (Figure 3b), but there are almost no changes of the XRD peaks locations in the temperature range (Figure 3a), why?

(2) Line 166 page 5, if the FDU-HOF-1 was dissolved in DMF solution, its crystalline structure would be broken, the discussion on the stability of the material seemed no meaning.

(3) Since the HOF materials are fragile and prone to collapse, how to keep its membrane strength in the ion separation application?

(4) The errors in checkcif file should be explained in the supporting information.

Round 2

Reviewer 2 Report

The authors have modified the manuscript and carefully considered the comments.